# Bio-Inspired Nanostructured Ti-6Al-4V Alloy: The Role of Two Alkaline Etchants and the Hydrothermal Processing Duration on Antibacterial Activity

**DOI:** 10.3390/nano12071140

**Published:** 2022-03-29

**Authors:** Richard Bright, Andrew Hayles, Jonathan Wood, Neethu Ninan, Dennis Palms, Rahul M. Visalakshan, Anouck Burzava, Toby Brown, Dan Barker, Krasimir Vasilev

**Affiliations:** 1Academic Unit of STEM, University of South Australia, Mawson Lakes, Adelaide, SA 5095, Australia; richard.bright@unisa.edu.au (R.B.); andrew.hayles@unisa.edu.au (A.H.); jonathan.wood@mymail.unisa.edu.au (J.W.); neethun.ninan@gmail.com (N.N.); dennis.palms@unisa.edu.au (D.P.); anouck.burzava@unisa.edu.au (A.B.); 2Division of Biomaterials and Biomechanics, Department of Restorative Dentistry, School of Dentistry, Oregon Health and Science University, Portland, OR 97201, USA; madathip@ohsu.edu; 3Corin Australia, Sydney, NSW 2153, Australia; tobydbrown@gmail.com (T.B.); dan.barker@coringroup.com (D.B.); 4College of Medicine and Public Health, Flinders University, Bedford Park, SA 5042, Australia

**Keywords:** antibacterial, biofilm, biomimetic, nanostructure, titanium alloy, implant associated infection

## Abstract

Inspired by observations that the natural topography observed on cicada and dragonfly wings may be lethal to bacteria, researchers have sought to reproduce these nanostructures on biomaterials with the goal of reducing implant-associated infections. Titanium and its alloys are widely employed biomaterials with excellent properties but are susceptible to bacterial colonisation. Hydrothermal etching is a simple, cost-effective procedure which fabricates nanoscale protrusions of various dimensions upon titanium, depending on the etching parameters used. We investigated the role of etching time and the choice of cation (sodium and potassium) in the alkaline heat treatment on the topographical, physical, and bactericidal properties of the resulting modified titanium surfaces. Optimal etching times were 4 h for sodium hydroxide (NaOH) and 5 h for potassium hydroxide (KOH). NaOH etching for 4 h produced dense, but somewhat ordered, surface nanofeatures with 75 nanospikes per µm^2^. In comparison, KOH etching for 5 h resulted sparser but nonetheless disordered surface morphology with only 8 spikes per µm^2^. The NaOH surface was more effective at eliminating Gram-negative pathogens, while the KOH surface was more effective against the Gram-positive strains. These findings may guide further research and development of bactericidal titanium surfaces which are optimised for the predominant pathogens associated with the intended application.

## 1. Introduction

Worldwide, there is a growing demand for implantable medical devices, in part due to an aging population made possible by advancements in medicine and technology [1]. Titanium and its alloys are the materials of choice in the field of orthopaedics, mainly due to their biocompatibility, excellent strength, and corrosion resistance [2]. While there is a high success rate for total joint replacement surgeries, implant failure is a pervasive and unrelenting threat. The primary cause of implant failure is due to bacterial contamination, and it is estimated that 1.5–2.5% of orthopaedic implants become a site of infection [3]. Implant-associated infection (IAI) is a devastating complication which is associated with severe morbidity and a mortality rate between 2.7 and 18% [4,5,6]. Evidence suggests that the presence of an implant reduces the minimal inoculum of bacteria required to cause infection by a factor of greater than 100,000 [7,8]. IAI can occur through multiple paths of pathogenesis. In one setting, bacteria residing on the patient’s skin can translocate to the implanted device during surgery [9,10]. Alternatively, infection can arise from haematogenous transfer, whereby bacteria at distal sites of infection travel through the bloodstream and encounter the implanted device [11,12]. In another mode of pathogenesis, contiguous spread from infected tissue (e.g., trauma, pre-existing osteomyelitis, and soft tissue lesions) act as a reservoir for pathogens to transfer to the implanted device [9,13]. Regardless of the route of pathogenesis, bacterial attachment, and colonisation of the implant surface, IAI is the unfortunate outcome. Bacteria residing on the implanted device strongly attach to the surface and proliferate to form a biofilm in a sequence of stages. Biofilm formation can be characterised by 3 broad phases: attachment, maturation, and dispersal [14]. Primary attachment is reversible and involves van der Waals and electrostatic forces [15]. Attachment is gradually strengthened by the presence of proteins, known as adhesins, which facilitate covalent linkage of the cell to the surface. The maturation phase of biofilm formation is marked by the production and secretion of extracellular polymeric substance (EPS), which contains polysaccharides, proteins, lipids, extracellular DNA, and quorum sensing molecules [16,17]. EPS is multifunctional, acting as a structural support, a medium for the exchange of molecules, and a defence against antimicrobial compounds and phagocytic cells. Once a biofilm has progressed through the maturation phase, it is exceedingly difficult to treat with antibiotics, and mature biofilms are known to be up to 1000× more resistant to antibiotics compared to their planktonic counterparts [18,19,20]. Due to this, IAI usually requires surgical intervention to remove or replace the implanted device [21]. When fully matured, the biofilm acts as a reservoir of cells which can be readily dispersed into the neighbouring environment, potentially causing further infection in sites around the body, or lethal septicaemia [22,23,24].

To tackle the burden of IAI, much research has focused on the modification of implanted biomaterials to grant them with anti-infective properties. For example, surfaces can be coated with antimicrobial peptides or drug-eluting compounds [25,26,27,28]. An alternative strategy is to modify the nanoscale topography of the surface itself to generate protruding structures which are hostile to bacteria but accommodating to host cells. Bacteria attaching to nanoscale protrusions have their cell membrane perturbed and penetrated, and this mechanical interaction is associated with induced oxidative stress and cell death [29]. The bactericidal effect of nanoscale protrusions was first observed on cicada wings by Ivanova and colleagues [30] and has since served as inspiration for the fabrication of a new generation of anti-infective surfaces. Bioinspired nanoscale protrusions have been successfully fabricated on silicon [31] and titanium [32,33] using a range of techniques. Hydrothermal etching is one such technique which has been used to modify titanium surfaces. It is attractive due to its simplicity, cost-effectiveness, and potential for large-scale manufacture [34]. The process of hydrothermal etching involves submerging a sample in an alkaline solution at high temperatures to form an oxide layer with nanoscale architecture [35]. The morphology of the resultant nanostructure is influenced by the combination of fabrication parameters used in the process including primarily etching duration, temperature, and alkaline etchant.

The purpose of this study is to examine the role of etchant type and processing time on the morphology of the resultant surfaces and their antibacterial capacity. Potassium hydroxide (KOH) and sodium hydroxide (NaOH) were used as etchants. Etching time was 1, 3, 4, and 5 h. The modified surface was challenged with three clinically relevant pathogens: *Staphylococcus aureus* (*S. aureus*), *Escherichia coli* (*E. coli*), and *Pseudomonas aeruginosa* (*P. aeruginosa*). The Gram-positive coccus *S. aureus* was chosen due to its ubiquitous association with IAIs [9], and well-documented resilience to mechanical killing [36] combined with its capacity to develop antibiotic resistance [37]. *E. coli* and *P. aeruginosa* were chosen as representative Gram-negatives which frequently appear in implant infections [38,39]. *P. aeruginosa* rods are typically longer than *E. coli* rods (up to 5 µm and up to 2 µm, respectively) [40,41] which makes them good examples to study the relationship between pathogen morphological characteristics and nanostructure topographical dimensions.

## 2. Materials and Methods

### 2.1. Fabrication of Hydrothermally Etched Ti-6Al-4V

Ti-6Al-4V discs (10 mm in diameter, 3 mm in height, and a surface area of 0.78 cm^2^) were received polished at a Ra of 0.5 µm (Hamagawa Industrial (M) SDN BHD, Kedah, Malaysia). To create the nanostructured surface, the discs were hydrothermally etched at 150 °C, using either 1 M KOH or 1 M NaOH in a stainless-steel reactor (Parr Instrument Company, Moline, IL, USA). Next, the reactors were cooled, and the samples were cleaned in ultrapure water. Ti-6Al-4V discs were then dried, followed by heat treatment inside an oven and allowed to cool down overnight. The as-received titanium alloy discs were used as controls (AR-Ti) and hydrothermally etched discs, using 1 M NaOH and 1 M KOH aqueous etching solutions, were then cleaned and sterilised at 121 °C for 20 min prior to use. Samples were fabricated using both alkaline solutions, (1 M KOH and 1 M NaOH) with etching times of 1, 3, 4, and 5 h.

### 2.2. Characterisation of the Surface Nanotopography

Scanning electron microscope (SEM) characterisation was performed on samples fabricated using both alkaline etchant solutions (1 M KOH and 1 M NaOH) with etching times of 1, 3, 4, and 5 h. The morphology and distribution of nanostructures upon the surface of the titanium substrate were analysed on a field emission gun scanning electron microscope Zeiss Merlin FEG-SEM (Zeiss, Jena, Germany), equipped with a secondary electron (SE) detector, at 2 KV with magnification from 5–50 K. The stage was tilted at 45 degrees for imaging of the nanostructures, and orthogonal when analysing density of individual nanostructures and imaging bacteria on the surfaces. The height of the nanostructures was determined by the linear distance between a basal plane and the highest point of each spike (*n* = 20), whereas the diameter was measured at mid-height in parallel orientation with the basal plane, and a correction factor of x/cos (45°) to amend dimensional distortion during linear measurements, where x equals the length of nanostructures. Spike height and diameter at mid height were expressed as mean ± standard deviation, and *n* = 5. The spacing between nanostructures was determined from four zero-degree tilted SEM images, considering the nanostructure tips in a 25 µm^2^ area and presented as mean ± standard deviation. Density was calculated by counting spike tip and expressing as nanostructures per μm^2^. The nanostructures dimensions were determined using ImageJ software version 1.53f51 (NIH, Bethesda, MD, USA).

### 2.3. Atomic Force Microscopy of 1M NaOH-Etched for 4 h (NaOH-4h) and 1M KOH-Etched for 5 h (KOH-5h) Samples

Atomic force microscopy (AFM) was used to acquire 25 µm^2^ images in air using a JPK NanoWizard III with instrument-specific software v5. An NT-MDT NSG03 silicon nitride cantilever with a conical tip rated by the manufacturer at a radius less than 10 nm and a half side angle of 18° was used to perform tapping mode on annealed AR-Ti and the optimal bactericidal etched samples, NaOH-4h and KOH-5h. Initial calibration of the cantilever on a glass microscope slide derived a normal spring constant of 1.8 N/m at just off the first resonant frequency of 84.5 kHz. Scanning parameters over a scan rate of 0.7 Hz were at a set point of 22 nm and a drive amplitude of 1.24 Volts. Roughness values were calculated, and images acquired using Gwyddion data analysis software v2.54 (http://gwyddion.net/, accessed on 30 August 2021).

### 2.4. Surface Analysis by X-ray Photoelectron Spectroscopy (XPS) 

Chemical composition of the upper 10 nm layer of AR-Ti, NaOH-4h, and KOH-5h were analysed using X-ray photoelectron spectroscopy (XPS). XPS survey spectra were collected using a Kratos AXIS Ultra DLD spectrometer (Kratos Analytical Ltd., Manchester, UK) equipped with a magnetically confined charge compensation system, with monochromatic AlKα radiation (hν = 1486.7 eV). The sample area analysed was 300 μm × 700 μm at a pass energy of 160 eV. Data analysis was performed with CasaXPS software (Casa Software Ltd., Teignmouth, UK). All binding energies were referenced to the low energy, aliphatic C 1 s peak at 285.0 eV.

### 2.5. Contact Angle of NaOH-4h and KOH-5h

Surface wettability was evaluated for AR-Ti, NaOH-4h, and KOH-5h using contact angle evaluated by the sessile drop method using a contact angle goniometer model RD-SDM02 (RD Support, Edinburgh, UK). The contact angle from probe liquid, ultrapure water (4 μL) was measured by a tangent fitting method using plugin Contact_Angle.jar for Image J software version 1.53f51 (NIH, Bethesda, MD, USA).

### 2.6. Bacterial Cultures

*Escherichia coli* ATCC 11303 (*E. coli*), *Pseudomonas aeruginosa* ATCC 15692 (*P. aeruginosa*), and *Staphylococcus aureus* ATCC 25923 (*S. aureus*) were cultured in tryptic soy broth (TSB; Oxoid, ThermoFisher, Waltham, MA, USA) and incubated overnight at 37 °C. Prior to starting bacterial experiments, it was determined that an optical density of 1 measured at 600 nm (OD_600_) was approximately 1 × 10^9^ CFU/mL

### 2.7. LIVE/DEAD BacLight Bacterial Viability

To compare the different etching times (1, 3, 4, and 5 h) and etchants (NaOH and KOH), overnight cultures of bacteria were adjusted to a concentration of 1 × 10^6^ CFU/mL, and 1 mL added to wells containing titanium discs (AR-Ti, NaOH, and KOH-etched samples) in triplicate, and incubated for 20 h at 37 °C in a humid chamber. Next, the samples were stained with LIVE/DEAD^®^ BacLight™ Bacterial Viability Kit (ThermoFisher Scientific, Waltham, MA, USA) as per the manufacturer’s instructions and analysed by an Olympus FV3000 laser confocal microscope (CLSM; Olympus, Tokyo, Japan). Samples were inverted onto a glass coverslip and four randomly selected regions were imaged. SYTO 9 and propidium iodide fluorescence excitation/emissions were monitored at 480/500 nm and 490/635 nm, respectively. The viability was determined using ImageJ software v1.53a (NIH, Bethesda, MD, USA).

### 2.8. Bacteria–Nanotopgraphy Interaction by SEM

Overnight cultures of *E. coli*, *P. aeruginosa*, and *S. aureus* incubated on AR-Ti, NaOH-, and KOH-etched samples were fixed for 1 h with 4% paraformaldehyde, 1.25% glutaraldehyde, and 4% sucrose in PBS. Following fixation, they were washed in PBS, followed by dehydration in an ascending ethanol order from 50% (*v*/*v*) to absolute ethanol and further dried using hexamethyldisilazane (HMDS; Sigma-Aldrich, St. Louis, MI, USA). After the dehydration process, the discs were mounted on aluminium stubs using double-sided carbon tape, sputter-coated with 2 nm platinum, and examined using a Zeiss Merlin FEG-SEM (Zeiss, Jena, Germany).

### 2.9. Cytocompatibility Analysis of NaOH-4h and KOH-5h Surfaces

Primary human derived dermal fibroblast cells (HDF) were seeded on AR-Ti, NaOH-4h, and KOH-5h samples in a 48 well plate. Cultures were maintained in Dulbecco’s modified Eagle medium (DMEM; Life Technologies, Carlsbad, CA, USA) supplemented with 10% *v*/*v* fetal bovine serum (FBS; Life Technologies, Carlsbad, CA, USA) and 1% (*v*/*v*) penicillin/streptomycin (Life Technologies, Carlsbad, CA, USA). Tissue culture plate (TCP) and AR-Ti discs were used as positive controls. HDF cells were seeded at a density of 2.5 × 10^4^ cells/well. Short-term cytocompatibility was assessed using the resazurin assay (Resazurin sodium salt, R7017, Sigma Aldrich, St. Louis, MI, USA). A stock solution of 100 µg/mL resazurin was prepared in phosphate buffered saline (PBS) and diluted to a final working concentration of 10 µg/mL. Briefly, the cells were incubated on TCP, AR-Ti, NaOH-4h, and KOH-5h for 48 h at 37 °C and 5% CO_2_, after which the culture medium was replaced with 250 μL media containing 10% resazurin of the stock solution and incubated for 1 h. Next, 100 µL of the supernatant was transferred into a 96 well plate and the fluorescent intensity was recorded using a plate reader. Resazurin is a fluorescent assay that detects cellular metabolic activity. The blue nonfluorescent resazurin reagent is reduced to highly fluorescent resorufin by dehydrogenase enzymes in metabolically active cells. The resorufin formed in the assay was quantified by measuring the fluorescent intensity using a plate reader (Ex = 530–570 nm, Em = 590–620 nm). Percent of cell viability is calculated using the following formula, $\frac{\mathrm{Fluorescent}\;\mathrm{intensity}\;\mathrm{of}\;\mathrm{treated}\;}{\mathrm{Fluorescent}\;\mathrm{intensity}\;\mathrm{of}\;\mathrm{control}\;}\times 100$, normalised to TCP control. To investigate changes in HDF cell morphology after 48 h incubation upon the surfaces, Phalloidin (Alexa Fluor 488 phalloidin, A12379, Thermo Fisher Scientific, Waltham, MA, USA) was used to stain actin and DAPI (4′,6-Diamidino-2-Phenylindole, Dihydrochloride, D1306, Thermo Fisher Scientific, MA, USA) to visualise the nucleus following recently published work [42]. Samples were washed in PBS, inverted onto a glass coverslip, and imaged by an Olympus FV3000 confocal laser scanning microscope (CLSM; Olympus, Tokyo, Japan).

### 2.10. Statistical Analysis

Statistical analysis for surface parameters and viability was carried out using one-way ANOVA or Student’s *t*-test followed by post hoc analysis using a Tukey’s multiple comparisons test. To compare nanostructure topographical dimensions, a Student’s *t*-test was performed. Statistical analysis was performed using GraphPad Prism version 8.3.0 (GraphPad Software, San Diego, CA, USA., www.graphpad.com, accessed on 8 December 2021). All experiments were undertaken in triplicate and reported as mean and standard deviation. In all experiments, significance was set at *p* < 0.05.

## 3. Results

### 3.1. Morphology and Dimensional Analysis of NaOH- and KOH-Etched Samples

A range of surface nanostructures were created on titanium alloy discs using hydrothermal etching with either NaOH or KOH aqueous solutions. Surfaces were etched for 1, 3, 4, or 5 h to investigate the changes in geometrical and morphological characteristics to investigate the nanostructure evolution with time. Surfaces were then annealed to improve corrosion resistance, alloy plasticity, abrasion resistance, and potential for bone ongrowth, and reduce surface stress developed during fabrication [43,44]. The SEM micrographs presented in Figure 1 show the formation of nanostructures, resembling those found in nature such as on dragonfly wings (Appendix A) [45,46], observed on both NaOH- (Figure 1A–E) and KOH-etched surfaces (Figure 1F–J). Although displaying an overall similar “spike-like” shape, architectural differences could be observed between the two etching solutions tested. Nanostructures on NaOH-etched surfaces appeared more densely packed, straighter (Figure 1A–E), and shorter than those formed on KOH-etched surfaces (Figure 1E–J). The nanostructures formed on the KOH-etched samples appeared to be arranged in a hierarchal disordered format (Figure 1F–J) [47].

The SEM micrographs were used to measure spike height, diameter at mid-height, spacing between, and density per µm^2^. The results are presented in Figure 2A and Appendix A for the NaOH-etched samples and Figure 2B and Appendix A for the KOH-etched samples. The spike height significantly increased as the etching time increased between 1 and 5 h for both NaOH- and KOH-etched surfaces (*p* < 0.0001). The nanostructure diameter at mid-height, followed an upward trend for NaOH, with a significant increase between 1 h and 3 h etching times (30 ± 4 nm and 94 ± 33 nm, respectively, *p* < 0.0001), and 1 h and 5 h etching times (30 ± 4 nm at 1 h and 83 ± 30 nm, respectively, *p* < 0.0001). However, the nanostructures’ diameter at mid-height remained almost constant from 1 to 5 h in the case of KOH-etching (65 ± 10 nm and 83 ± 32 nm, respectively, *p* = 0.02). The spacing between nanostructures remained unchanged for etching times on the NaOH-etched surfaces (1 h and 5 h etching times, 185 ± 41 nm, and 224 ± 55 nm, respectively, *p* = 0.02), and increased on KOH-etched surfaces (1 h and 5 h etching times, 330 ± 83 nm and 544 ± 150 nm, respectively, *p* < 0.0001). This indicates that there was no new nanostructure formation over time, only evolution of those that had already formed at the beginning of the hydrothermal process, the main outcome being the increase in nanostructure height. The density of the nanostructures was lower on the KOH-etched surface (8 ± 2 spikes/ µm^2^ at 5 h etching time) compared to NaOH-etched surface (32 ± 9 spikes/µm^2^ at 5 h etching time). 

### 3.2. Analysis of Bacterial Morphology Using SEM

The interactions between the nanostructures and *E. coli*, *P. aeruginosa*, and *S. aureus* on the surfaces generated by NaOH and KOH hydrothermal etching for 1, 3, 4, and 5 h are shown in Figure 3 and Figure 4. We observed morphologically disturbed cells on all treated surfaces (highlighted by yellow arrows). We observed the highest proportions of dead cells on NaOH-4h and KOH-5h. Overall, the two Gram-negative species were more frequently observed to be damaged and flattened against the surfaces compared to the Gram-positive *S. aureus*. This is somewhat expected as Gram-positive bacteria possess a thicker peptidoglycan layer which confers greater structural support compared to the relatively fragile Gram-negative species [34]. However, despite the greater rigidity of *S. aureus*, a high proportion of damaged cells were observed on the KOH-5h surface (Figure 4).

### 3.3. Bacterial Analysis by Live/Dead Assay

Live/Dead fluorescence analysis for the target bacterial species is shown in Figure 5 and Figure 6, with the percentage viabilities for each pathogen on each surface shown in Figure 7. The two Gram-negative species were notably more vulnerable to nanostructure induced cell death compared to *S. aureus*. For *P. aeruginosa*, viability ranged from 3.5 ± 2.0% to 6.7 ± 2.7% on the NaOH-etched surfaces. On the KOH-etched surfaces, *P. aeruginosa* viability ranged from 4.6 ± 0.8 to 16.7 ± 1.0%. The most effective surface against *P. aeruginosa* was NaOH-4h, which reduced viability to 3.5 ± 2.0%. For *E. coli*, viability ranged from 20.7 ± 3.6% to 32.4 ± 2.5% on the NaOH-etched surfaces. On the KOH-etched surfaces, *E. coli* viability ranged from 38.1 ± 4.0% to 60.3 ± 4.1%. The most effective surface against *E. coli* was NaOH-4h, which reduced viability to 20.7 ± 3.6%. For the Gram-positive *S. aureus*, viability ranged from 45.2 ± 2.0% to 91.1 ± 7.0% on the NaOH-etched surfaces. On the KOH-etched surfaces, the *S. aureus* viability ranged from 38.5 ± 4.8% to 73.2 ± 6.3%. The most effective surface against *S. aureus* was KOH-5h, which reduced *S. aureus* to 38.5 ± 4.8%.

Based on the above data, the etching duration which produces nanotopography with the highest bactericidal activity for NaOH was 4 h, whereas for KOH it was 5 h. Additionally, it can be noted that the NaOH-4h surface was more effective at eliminating the two Gram-negative pathogens, while the KOH-5h surface was more effective at eliminating the Gram-positive *S. aureus* [48]. We therefore selected these two etching parameters for further surface property characterisation to understand their differing bactericidal activity.

### 3.4. Characterisation of the NaOH-4h and KOH-5h Samples

To find a potential explanation for the observed bactericidal selectivity between NaOH-4h and KOH-5h, we first compared the spike dimensions of the fabricated nanostructures (Table 1). We found no statistical difference between the height and diameter at mid-height of the nanostructures (*p* = 0.81 and *p* = 0.41, respectively). However, although the spike heights were not significantly different, the KOH-5h shows a much greater range in measured nanostructure height (175 nm) compared to the NaOH-4h (80 nm). This irregularity of spike heights can be clearly observed in both the SEM and AFM micrographs (Figure 1 and Figure 8, respectively). The NaOH-4h surface had significantly lower structure spacing (*p* = 0.02) and greater spike density (*p* = 0.0001) when compared to the KOH-5h surface.

Elemental composition detected on the AR-Ti, NaOH-4h, and KOH-5h surface using XPS is presented in Figure 8A. There was an increase in oxygen concentration of 5.6% on the NaOH-4h surface and 7.6% on the KOH-5h surface compared to the AR-Ti control. This is consistent with thickening of the oxide layer (TiO_2_) occurring during the hydrothermal etching process. Furthermore, the thicker oxide layer also resulted in a reduction in the surface concentration of vanadium and aluminium as XPS sampling depth is limited to the outermost 10 nm of the surface. This may prove to be advantageous for implantable biomaterials as wear particles and leaching of vanadium and aluminium from Ti-6Al-4V have raised concerns due to their potential toxicity and DNA damaging effects [49,50,51]. Additionally, small levels of sodium and potassium were incorporated within the NaOH-4h and KOH-5h surface, respectively, consistent with the use of sodium and potassium as cations for the alkaline treatment.

To compare the hydrophilicity of samples before and after hydrothermal etching, water contact angles (WCA) were measured. The WCA for NaOH-4h surface (7.6° ± 1.2°) and KOH-5h (9.7° ± 1.1°) was significantly lower than that for the AR-Ti surface (76.0° ± 5.0°), *p* < 0.0001, indicative of significantly increased hydrophilicity (Figure 8B). There was no significant difference in water contact angle between the NaOH-4h and KOH-5h surface treatments (*p* > 0.05). Such superhydrophilic behaviour is the result of the combination of surface roughness at the nanoscale and a hydrophilic TiO_2_ surface layer, consistent with the Wenzel and Cassie Baxter theory [52,53,54]. An increase in hydrophilicity is an imperative characteristic of the hydrothermally etched surface since the phenomenon is generally associated with improved biocompatibility and enhanced bactericidal properties [55,56].

The surface area and roughness measurements calculated from the AFM analysis can be found in Appendix A. AFM images confirmed marks due to the polishing process on the AR-Ti surface (Figure 8C). Nanostructures were observed on the NaOH-4h and KOH-5h samples (Figure 8D,E, respectively); however, fine details of nanostructures such as those seen by SEM were not observed by AFM. It should be noted that the tip convolution effect resulting from the cantilever geometry and scan velocity reduced the measured surface roughness values, particularly towards nanostructures with comparable magnitude to the tip radius [57]. AFM analysis for NaOH-4h and KOH-5h showed greater roughness and surface area compared to the AR-Ti surface. The root mean square (RMS), arithmetic roughness average (Ra), and surface area (SA) calculated from the AFM images for the AR-Ti surface were 10.1 nm, 6.6 nm, and 25.2 µm^2^ respectively, whereas for the NaOH-4h surface, these parameters were 88.4 nm, 71.4 nm, and 79.9 µm^2^, respectively. Compared to the NaOH-4h surface, the KOH-5h surface had a similar RMS value of 88.5 nm, and a decrease in both Ra and SA (61.5 nm and 50.3 µm^2^, respectively, Figure 8F). The relative surface area measurements between NaOH-4h and KOH-5h are consistent with the measured structure dimensions and the greater surface area of NaOH-4h is reflective of more densely packed nanospikes compared to KOH-5h.

Based on the above data, it can be noted that there are only two considerable differentiating characteristics between the NaOH-4h and KOH-5h surfaces. The first is the nonuniformity of nanostructures length on KOH-5h, which creates a chaotic array of spikes of different heights, reminiscent of the dragonfly wing (Appendix A). In contrast, the nanospikes observed on NaOH-4h, were relatively uniform and had similar heights. The chaotic arrangement of spikes with nonuniform heights on the KOH-5h surface may therefore allow it to pierce bacteria from multiple angles rather than just from the bottom. The other perhaps more important differing characteristic is the spacing between nanospikes. On the NaOH-4h surfaces, the spikes are densely packed with a mean of 75 spikes per μm^2^, compared to the sparser KOH-5h surfaces with a density of approximately 8 spikes per μm^2^. These data suggest that the NaOH-4h nanoarchitecture supports bactericidal activity against Gram-negative rods, while a KOH-5h nanoarchitecture supports a high bactericidal efficacy of Gram-positive cocci. This may prove to be an important finding which could guide research and the development of nanostructures with specific functional outcomes. In the case of implantable biomaterials in the orthopaedic field, the KOH-5h surface may be preferred over the NaOH-4h surface due to its spike density favouring bactericidal activity against Gram-positive cocci, such as the clinically relevant *S. aureus*. As *S. aureus* is one of the most predominant bacterial pathogens in orthopaedic implant infections, implant manufacturers may benefit from treating titanium implants with the KOH-5h procedure described in the present report.

### 3.5. In Vitro Short-Term Cytocompatibility

The cytoskeleton and nuclei of HDF cells on the nanostructured surfaces (AR-Ti, NaOH-4h and KOH-5h) incubated for 48 h are shown in Figure 9. The HDF cells incubated on the AR-Ti surface exhibited polarised phenotype manifested by an elongate cell body (Figure 9A), whereas the cells incubated on NaOH-4h and KOH-5h surfaces were somewhat less polarised, with cells showing spreading upon the surfaces due to the interaction with the nanostructures [58]. This suggests the HDF cells incubated on these surfaces exhibit strong adhesion [59]. Cell viability determined via the resazurin method is shown in Figure 9C. The HTE-treated surface did not cause any cytotoxicity as the viability of the cells was 84.6 ± 16.4% on the NaOH-4h and 87.5 ± 16.9 on the KOH-5h, compared to the AR-Ti 82.1 ± 7.5%, *p* > 0.05 (Appendix A). These results indicate good viability and cell attachment on the nanostructured surfaces, indicating adequate cytocompatibility and biocompatibility.

## 4. Conclusions

We investigated the role of alkaline etchants and hydrothermal processing duration on the resultant surface nanotopography outcomes for titanium alloy discs. Nanostructured surfaces were challenged with three clinically relevant pathogens to determine their bactericidal potential. Our bacteriological analysis allowed us to narrow down the etching parameters to two favourable processing conditions involving either NaOH-etching for 4 h or KOH-etching for 5 h. NaOH-4h treatment exhibited spikes that are densely packed with a mean of 75 spikes per μm^2^, compared to the sparser KOH-5h with a density of approximately 8 spikes per μm^2^ suggesting that a greater density of spikes supports bactericidal activity against Gram-negative rods, while a sparser nanostructures array supports killing of Gram-positive cocci. Short-term in vitro cytocompatibility study using HDF cells incubated on NaOH-4h and KOH-5h surfaces showed no unfavourable effects on morphology, viability, and proliferation. Further research is required to fully understand the mechanisms supporting the antibacterial properties of the nanostructured surface. However, this study highlights that with appropriate optimisation, substrates with selective antibacterial activity could be potentially achieved. 

## Figures and Tables

**Figure 1 nanomaterials-12-01140-f001:**
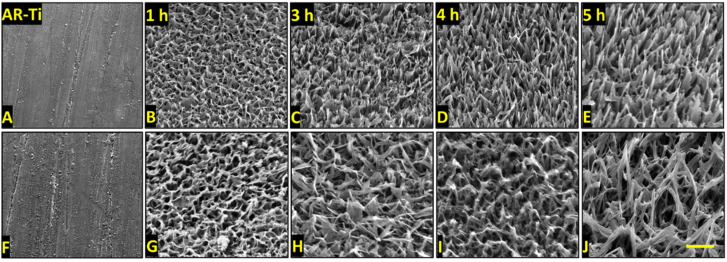
SEM micrographs showing the morphology and distribution of nanoarchitectures obtained after etching the surface for AR-Ti, 1, 3, 4, and 5 h using either NaOH (**A**–**E**) and KOH (**F**–**J**) aqueous-based etching solutions. Images were acquired at 50,000 magnification with 45° stage tilt, scale bar bottom left panel represents 500 nm.

**Figure 2 nanomaterials-12-01140-f002:**
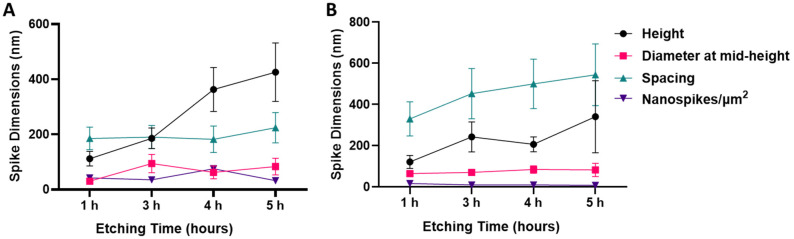
Nanostructures’ dimensions for NaOH-etched samples (**A**) and KOH-etched titanium samples (**B**) measured from four etching times (1, 3, 4, and 5 h). Mean ± SD. Details of nanostructures height, diameter at mid-height, spacing and density can be found in Appendix A.

**Figure 3 nanomaterials-12-01140-f003:**
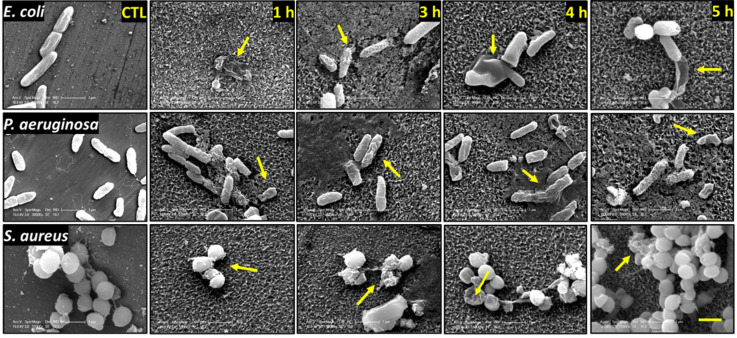
SEM micrographs of *E. coli*, *P. aeruginosa*, and *S. aureus* following 20 h incubation on the NaOH-etched samples (1, 3, 4, and 5 h etching time). Yellow arrows indicate damaged bacteria. SEM micrographs were acquired at 50,000 magnification, and scale bar bottom right image represents 1 µm.

**Figure 4 nanomaterials-12-01140-f004:**
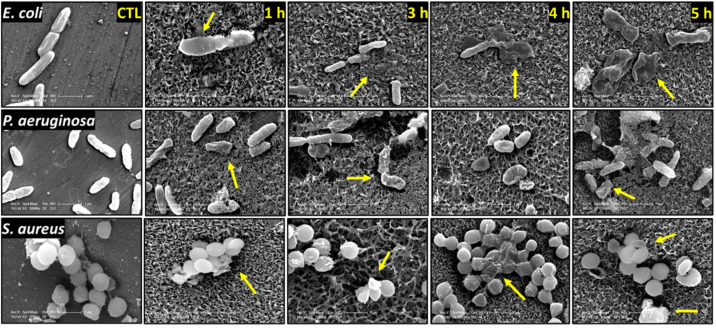
SEM micrographs of *E. coli*, *P. aeruginosa*, and *S. aureus* following 20 h incubation on the KOH-etched samples (1, 3, 4, and 5 h etching time). Yellow arrows indicate damaged bacteria. SEM images were acquired at 50,000 magnification. Scale bar bottom right image represents 1 µm.

**Figure 5 nanomaterials-12-01140-f005:**
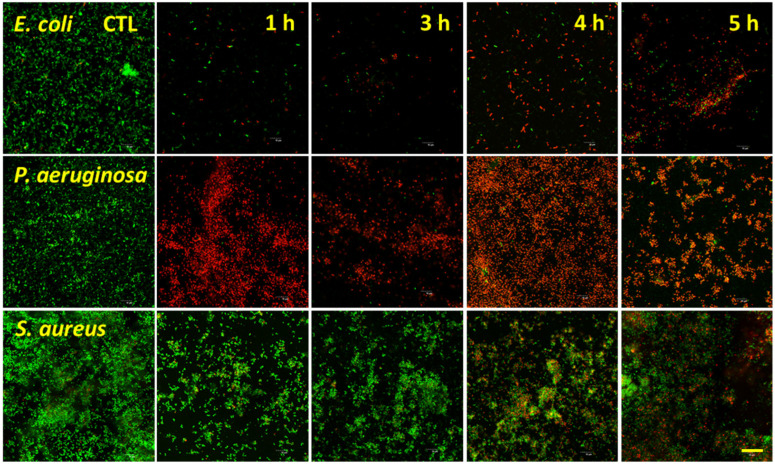
Confocal microscopy micrographs of *E. coli*, *P. aeruginosa*, and *S. aureus* incubated on the surfaces of the titanium etched using 1 M NaOH for 1, 3, 4, and 5 h. Bacteria were inoculated on the etched samples at a concentration of 10^6^ CFUs and incubated for 20 h. Cells were stained with a LIVE/DEAD BacLight™ bacterial viability kit. Scale bar represents 20 µm.

**Figure 6 nanomaterials-12-01140-f006:**
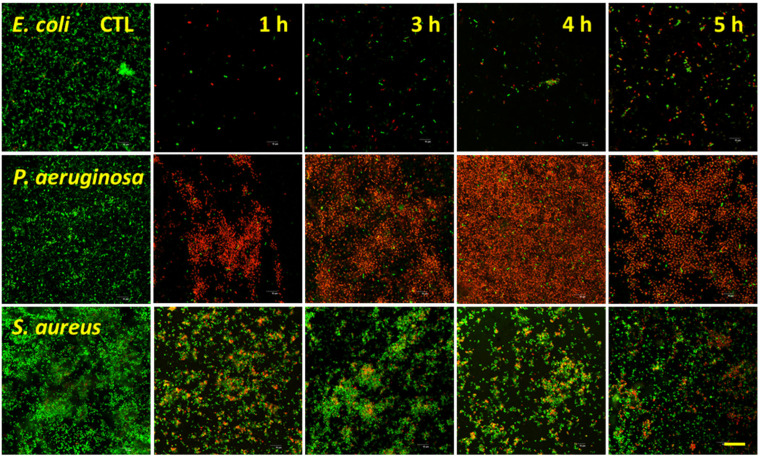
Confocal microscopy images of *E. coli*, *P. aeruginosa*, and *S. aureus* incubated on the surfaces of the titanium etched using 1 M KOH for 1, 3, 4, and 5 h. Bacteria were inoculated on the etched samples at a concentration of 10^6^ CFUs and incubated for 20 h. Cells were stained with a LIVE/DEAD BacLight™ bacterial viability kit. Scale bar represents 20 µm.

**Figure 7 nanomaterials-12-01140-f007:**
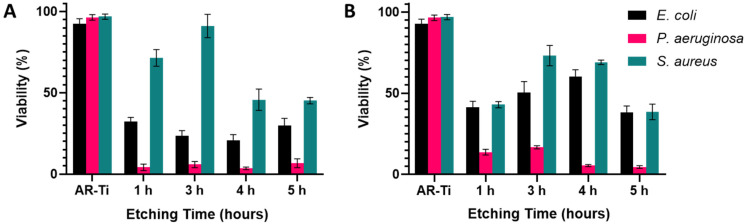
Percentages of viability observed across the three bacteria strains tested, after 20 h incubation on both NaOH-etched (**A**) and KOH-etched (**B**) for 1, 3, 4, and 5 h etching time, with initial seeding concentration of 10^6^ CFUs (mean ± SD, *n* = 3).

**Figure 8 nanomaterials-12-01140-f008:**
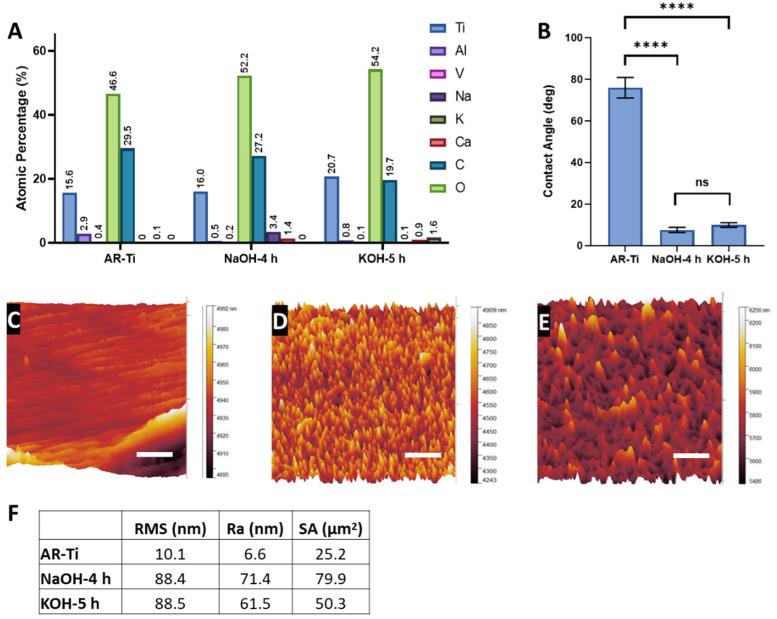
Surface analysis of NaOH-4h and KOH-5h, by XPS survey spectra (**A**), surface wettability (**B**) Mean ± SD, **** *p* < 0.001, ns = non-significant, and AFM 3D nanotopological features of a 25 µm^2^ region on AR-Ti, NaOH-4h, and KOH-5h samples (**C**–**E**), scale bars represent 1 µm. The root mean square (RMS), arithmetic roughness average (Ra), and surface area (SA) for AR-Ti, NaOH-4 h, and KOH-5 h samples, calculated using Gwyddion data analysis software from the AFM data (**F**).

**Figure 9 nanomaterials-12-01140-f009:**
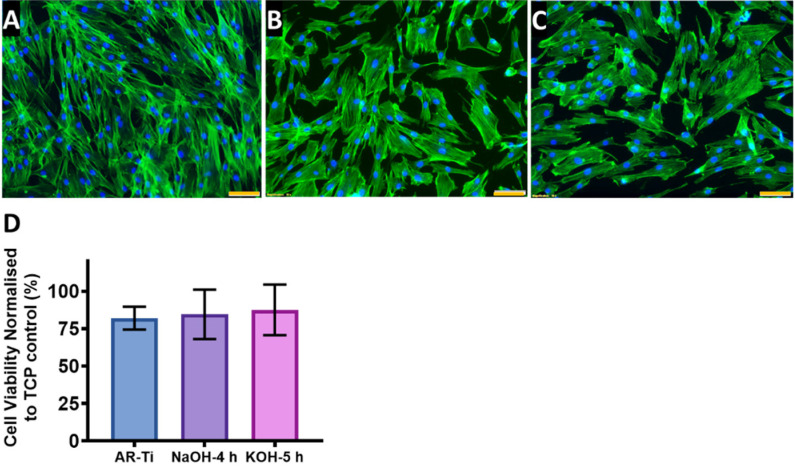
Confocal micrographs demonstrating cellular morphology of HDF stained with Phalloidin (green) and DAPI (blue) incubated on AR-Ti (**A**), NaOH-4h (**B**), and KOH-5h samples (**C**). Percentage cell viability normalised to TCP control (**D**), mean ± SD, *n* = 3 and scale bars in (**A**–**C**) represents 100 µm. *p*-values can be found in Appendix A.

**Table 1 nanomaterials-12-01140-t001:** Topography dimensional comparison between NaOH-4h and KOH-5h.

Parameter	NaOH-4h	KOH-5h	Significance (*p*)
Height (nm)	367 ± 80	340 ± 175	0.81
Diameter at mid-height (nm)	62 ± 23	83 ± 32	0.41
Spacing (nm)	182 ± 48	544 ± 150	0.02
Spike density (spikes/µm^2^)	75 ± 8	8 ± 2	<0.0001

## Data Availability

The datasets generated and/or analysed during the current study are available from the corresponding author on reasonable request.

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
