# Peer review of "Bio-Inspired Nanostructured Ti-6Al-4V Alloy: The Role of Two Alkaline Etchants and the Hydrothermal Processing Duration on Antibacterial Activity"

_nanomaterials, 2022, doi:10.3390/nano12071140_

Round 1

Reviewer 1 Report

The authors studied the role of two alkaline etchants and the hydrothermal processing duration on the resultant surface nanotopography outcomes for titanium alloy discs. Three clinically relevant pathogens were used to determine the bactericidal potential of the nanostructured surfaces. They correlated the process parameters with the antibacterial activity and defined the best combination of them.

The research design is accurate, the results well showed and the manuscript written using correct English language and style.

From my side, the manuscript can be accepted after minor revision.

Revision: Authors should check Table S1.

SD of the Density (spike/µm2) of NaOH-5h is missed in the Table S1.

“Diameter at mid-height” instead of “Diameter”.

There is no correspondence between some values in the text (lines from 246) and the same values in the Table S1.

For example in the text: “diameter at mid-height, for NaOH, with a significant increase between 1 h and 3 h etching times (21  ± 3 nm and 67 ± 23 nm respectively, p < 0.001), and 1 h and 5 h etching times (21 ± 3 nm at 1 h and 58 ± 22 nm respectively, p < 0.05)” while in the Table S1, NaOH 1 h (30  ± 4 nm), 3 h (62±23 nm) 5h (83 ± 30 nm).

Reviewer 2 Report

The research paper entitled Bio-Inspired Nanostructured Titanium Alloys: The Role of Process Parameters on Antibacterial Activity” deals with the antibacterial activity of the nanostructured surface of hydrothermally etched Ti6Al4V titanium alloy. The surfaces were prepared using either NaOH or KOH as etchants and the surface morphology was well characterised through various analysis techniques. A correlation was clearly established between the nanostructures features and the antibacterial activity against two Gram negative strains (P. aeruginosa and E. coli) and one Gram positive strain (S. aureus). The results reported suggest that rational design of nanostructured surfaces can assist the development of selective antibacterial bio-compatible materials that can tackle relevant biomedical challenges (but not only) such as implant-associated infections.

Given that the research carried out is highly relevant and systematically carried out, I recommend the publication, provided that first some minor issues and modifications are addressed:

  1. Some references and tables are referenced in the text but are not presented, notably table S3 (not present in the SI) and reference 59 (referenced in the text but not present in the refs list).
  2. I recommend a revision of the paragraph 2.2 (Characterisation of the surface nanotopography); the extrapolation of surface figures of merit from the images is explained in great detail, however this might create more confusion than actually assist the understanding of the method. I would suggest either for it to be reworded in a more concise way or in alternative to add in the SI a schematic of the surface structures and how they are treated. Also, lines 130-133 should be moved at the very beginning of the paragraph.
  3. Lines 207-212: the sentence should be reviewed, as something is clearly missing (maybe an “after” at the very beginning?).
  4. In general I find that there is a larger than necessary statement of the p-values along the text; this often makes the discussion more confusing, especially when such p-values are confronted with the error bars reported on the graphs. I will give one specific example: when reading that “the spike height significantly increased as the etching time increased for both NaOH and KOH etched surfaces (p<0.05)” (lines 245-246) and confronting this sentence with the big error bars on the corresponding graphs in Figure 2, there is clearly something off between the error and the p-value. By looking at the graphs and the error bars therein I would say that there is a significant increase in the spike height when comparing 1h etching vs 5h etching, whereas statistically there is no change between the same height for 4h and 5h etching with NaOH and 3h, 4h and 5h etching with KOH. I recommend a revision of the text from this point of view, everywhere there are comparison based on the p-values.
  5. Figure 1 as it is presented now has many too small panels, too small to clearly see the differences. Maybe a vertical orientation could increase the size and consequently the readability.
  6. From the micrographs presented Figure 3 I see a relatively comparable surface colonisation between the as received alloy and all the hydrothermally etched ones. I was expecting to see a lower colonisation degree instead, consequent to the nanostructures on the surface. Did the authors choose areas relative more colonised, to better show damaged cells, or the surface coverage of bacterial cells is representative of the whole surface? Could you comment on this? Also, the same comment for the figure orientation (see point 5) applies here.
  7. I suggest to move Figures 5 and 6 to the SI and leave here only the relevant information that is presented in Figure 7.
  8. Lines 370-371: I suggest moving them at the very beginning of the discussion of surface area and roughness extrapolated from AFM measurements.
  9. Figure 8 caption should be corrected with “5 x 5 µm2” (line 375) and the size of the scale bars on the AFM images should be added.

Reviewer 3 Report

Congratulations on your interesting article. In my opinion, the manuscript only needs a few small corrections (listed at the very end). My main complaint concerns the title:

- presented results relate to a specific alloy, not to quite a large group of titanium alloys;

- you haven't actually investigated the nanostructured alloy. I deal with nanostructured/nanocrystalline materials. It is generally accepted that the nanostructured alloys have ultrafine-grained microstructure, in case of Ti alloys, most often obtained by severe plastic deformation (SPD) methods. By the way, I suggest to add some information about the examined material - it is unclear whether the discs were cut from a bar;

- “Process Parameters” sounds too genera. I’d suggest telling the potential reader what the process is about.

In summary, the title should be worded as follows: “Bio-Inspired Nanostructured Surface of Ti-6Al-4V Alloy: The Role of Etching Parameters on (Its?) Antibacterial Activity”.

Other remarks:

- I strongly recommend to use the “Ti-6Al-4V” alloy designation, instead of “Ti6Al4V”;

- p. 6, Fig. 2 and p. 9, Fig. 7 – You should add a title of the X axes;

- p. 12, Fig. 9 – there is no information abut the scale bar.

Reviewer 4 Report

In the paper, the authors inspired by observations done on cicada and dragonfly wings by other researchers have sought to mimic these nanostructures onto biomaterials. The main goal was to reduce implant associated infections. For the studies, titanium and its alloys (Ti6Al4V) are usually used for biomaterials. They have excellent anti-corrosion properties but are susceptible to bacterial colonization.
For the studies, the alkaline heat treatment in NaOH and KOH solutions was performed in view of getting modified topographical, physical and bactericidal properties of the titanium surfaces. The NaOH surface appeared to be more effective at eliminating Gram-negative pathogens, while the KOH surface was more effective against the Gram-positive strains. These findings may be the basis to guide further research and development of bactericidal titanium surfaces. The authors suggest optimization for the predominant pathogens and future application.
The study method used in this work and the results obtained are worth of publication. The paper is well written, though there are some points noticed for corrections. First of all, the authors mention Table 2, and Table 3 in the text, but they have not included them in the manuscript (?). Figures and Tables noticed first time should be in bold. Space should be put between the value and untit, but „per cent” cannot be separtated (alike angle degree)!!
To sum up, this manuscript should be corrected before it is advised for publication in Nanomaterials
